# A Data-Driven DAE-CNN-BiLSTM-Attention Prediction Model for the State of Health of Lithium-ion Batteries

1st Can Zhang
*School of Electrical Engineering*
*Southwest Jiaotong University*
Chengdu, China
zhangcan@my.swjtu.edu.cn

2nd Yuanjiang Hu
*School of Electrical Engineering*
*Southwest Jiaotong University*
Chengdu, China
360832800@qq.com

3rd Deqing Huang
*School of Electrical Engineering*
*Southwest Jiaotong University*
Chengdu, China
elehd@home.swjtu.edu.cn

4th Jiaxin Fang
*School of Electrical Engineering*
*Southwest Jiaotong University*
Chengdu, China
2022210644@my.swjtu.edu.cn

5th Na Qin
*School of Electrical Engineering*
*Southwest Jiaotong University*
Chengdu, China
qinna@swjtu.edu.cn

*Abstract*—Accurately predicting health state of lithium-ion batteries is essential for their safety, reliability, and longevity. Predicting the State of Health (SOH) using health indicators is a proven effective method. However, real-world battery charge-discharge data is often noisy, particularly during capacity regeneration. To achieve accurate health state predictions, we extracted over ten health indicators and designed a hybrid model: DAE-CNN-BiLSTM-Attention. This model integrates the strengths of Convolutional Neural Networks (CNN) for local feature extraction,and Bidirectional Long Short-Term Memory networks (BiLSTM) for temporal dependency learning, the Attention mechanism for effective weight assignment, and Denoising Autoencoders (DAE) for restoring original data,enabling the network to better adapt to complex real-world environments.The adaptability and stability of the proposed model were validated using two public datasets: NASA and CALCE. Compared to other existing methods, the proposed model demonstrated superior performance, achieving mean absolute error (MAE) and root mean square error (RMSE) of 0.0154 and 0.0191, respectively.

*Index Terms*—State of Health(SOH), Lithium-ion Battery(LiB), convolution neural network(CNN), feature extraction, long short-term memory (LSTM)

## I. INTRODUCTION

To address environmental challenges and the fossil energy crisis, there is an urgent and vigorous development of clean energy sources as hydro,wind and nuclear power. Consequently, the issue of energy storage and utilization has become particularly critical. Lithium batteries,compared with other types of batteries, such as NiMH batteries,lead-acid batteries, offer higher energy density, lower self-discharge rates, and longer charge-discharge lifespans. These advantages have led to their widespread application, including , electric vehicles,portable electronic electronics, and energy storage systems [1].However, over time and with usage, batteries inevitably experience aging. This results in increased internal resistance, reduced usable capacity, and degraded performance, which can lead to battery leakage, localized short circuits, and potential safety hazards such as device malfunctions, shutdowns, or even overheating and explosions. Consequently, in critical applications, batteries are often replaced periodically to ensure safety, which inevitably leads to resource wastage. Battery Management Systems (BMS) are essential to ensure that batteries function safely, reliably and efficiently, with the health state being a core concern. Accurately predicting the SOH is vital for assessing battery aging, conserving resources, and ensuring batteries safety [2]- [4].

The health status of the battery (SOH) is a key indicator of its performance deterioration, and it quantifies the rate of battery aging by a percentage. As batteries age, the percentage gradually decreases, a phenomenon commonly described as a reduction in the total available capacity of the battery and an increase in the resistance. The SOH value directly reflects the current health of the battery, and the higher the value, the better the battery state. In order to estimate the SOH of battery accurately, researchers have developed a variety of monitoring technologies to monitor the voltage, current and temperature of battery in real time. In general, battery health is measured by the ratio of the maximum available capacity to the rated capacity of the battery [5]. This ratio can be used to predict the health of the battery and to advise users on when to replace it. Therefore, understanding the SOH of the battery is essential to extend its life and ensure safe operation. Through accurate monitoring and analysis of these parameters, manufacturers can adjust the strategy of battery management system to maintain the best performance and prolong the life of battery.This paper also adopts this definition, with the SOH defined as shown in Eq (1).

$$SOH = \frac{C_{\max}}{C_{\text{norm}}} \times 100\% \qquad (1)$$

where $C_{\max}$ and $C_{\text{norm}}$ are two key parameters, which represent the actual maximum capacity and the standard rated capacity of the battery.In the field of battery data-driven research, If the maximum available capacity of the battery falls below 70% of its initial value, it is usually regarded as a warning line, known as a failure threshold. Such a situation in high-speed rail(HSR) and electric vehicle(EV) batteries could mean serious aging or health problems that require timely attention and maintenance. According to the description of the relevant literature [6], the battery in this case should not be used for high load or long time application occasions to avoid potential safety risks.

Due to the complex operating environments of batteries, such as temperature variations and the internal chemical reactions within the battery, which introduce uncertainties, the time-varying and highly nonlinear characteristics of batteries make accurately predicting SOH a challenging research problem [7]. Currently,these technologies can be divided into two main categories: model-based and data-driven. Battery fault diagnosis technology based on model method is to predict the health of the system by extracting model parameters of the system. Through in-depth study and detailed analysis of the physical and chemical properties of the battery, the equivalent circuit model is constructed to accurately simulate the behavior of the battery. [8]- [9] or electrochemical models [10]. Typically, state observers are used to describe the degradation mechanisms between battery cycles [11], such as Kalman filters [12]- [13] and particle filters [15]. Although electrochemical models have relatively high accuracy, they rely on precise electrochemical impedance spectroscopy. On the other hand, equivalent circuit models are less satisfactory because they fail to capture the aging characteristics of the battery. Model-based approaches often involve ideal or empirical models that do not account for internal chemical reactions and aging mechanisms, making accuracy increasingly difficult to maintain over time [1]. Additionally, the physical and chemical parameter models of batteries are very complex, which imposes severe limitations due to measurement difficulties, robustness, dynamic accuracy, and poor adaptability.

Contrast the data-driven approach with the model-based approach, which does not require consideration of these parameters. Instead, they directly extract and analyze historical charge and discharge data from the battery, By using machine learning or deep learning techniques to dig deep into the rich information hidden in the data, the relevance of these characteristics to the state of health (SOH) is revealed. Examples include Support Vector Machines (SVM) [17], Backpropagation (BP) neural networks [16], Relevance Vector Machines (RVM) [18], and Bayesian networks [19]. However, considering the time dependency of battery degradation data, recurrent neural networks (RNNs) have shown superior predictive performance. Literature [11] has already suggested using RNNs for battery SOH prediction. LSTM as an upgraded version of RNNs [20], prevent issues like gradient explosion and perform exception-

ally well in sequence prediction. To connect the degradation data over time, some studies have used bidirectional LSTM networks [21]. In order to break through the defects of single network model, many research uses hybrid network technology to improve the prediction performance. For example, in cite b22, the LSTM network is used to predict battery life using the empirical mode decomposition (EMD) method. This decomposition method can capture the complex dependencies between different states within the battery, thus providing a more accurate model for battery life prediction. The article cite b23 shows how to combine a closed cycle cell (GRU) with a convolution neural network (CNN) to predict the working state of lithium-ion batteries (SOH). By combining the advantages of the two neural networks, this study aims to gain a more comprehensive understanding of the degradation process of battery performance and to assess its remaining service life (RUL). This interdisciplinary combination provides a new perspective for understanding and optimizing battery management systems. Study cite b24 estimates SOH and predicts RUL by constructing a hybrid network of LSTM and CNN (CNN-LSTM). This hybrid network combines the ability of two neural networks to process sequence data and image data, giving full play to their respective advantages to achieve higher prediction accuracy and efficiency.

In exploring the estimation of lithium-ion battery life (SOH), the researchers not only limited themselves to using different algorithms, but also looked at a range of health factors (HIs). Including constant current constant voltage scheme [25], Open Circuit Voltage (OCV) [26], Incremental Capacity (IC) curve peaks [27], cycle numbers [28], differential capacity [29], and differential voltage [30], which describe battery degradation. Using external characteristic parameters such as current, voltage and temperature as health factors [31], and the HIs closely related to attenuation are screened by Pearson correlation analysis.

Despite achieving good prediction results, most of the existing studies are based on the neural network (NN) training of hidden layer features in the dimensions of the weight is the same. However, each feature has a different effect on the SOH. Ignoring this factor will affect the accuracy of the prediction. Attention mechanisms, including channel attention (dimension attention), multiple attention cite b11, spatial attention and time attention cite b32.The attention mechanism improves the performance of the network model by dynamically focusing on the key information related to tasks. It can identify data points that are of greater importance in a particular context, thus facilitating the model's ability to understand and predict complex scenarios. Moreover, real-world data are often noisy, especially capacity regeneration process [33].

In summary, this paper proposes a novel hybrid network model. Initially, the data is denoised, followed by the extraction of local features using two convolutional layers. Long-term dependencies within the sequence are captured through a bidirectional LSTM, while a temporal attention mechanism focuses on each timestep in the sequence, assigning a weight to each timestep to emphasize important points and improve

the handling of sequential data. The main contributions of this paper are reflected in the following aspects:

1. The accuracy and feasibility of the DAE-CNN-BiLSTM-Attention model for SOH prediction were validated on two commonly used public datasets, NASA and CALCE.

2. Ten health indicators related to battery aging, including time, temperature, voltage, current, and internal resistance, were extracted. To avoid the interference of multiple factors, the top five most relevant health indicators for each battery were selected as network input features.

3. The model considered the battery capacity regeneration and the impact of real-world noise by incorporating a denoising step in the code, enhancing the model's robustness.

The following are the arrangements for the remainder of this article: Section 2 describes the methods used, including feature extraction and the proposed network model. Section 3 validates the model's effectiveness with actual battery data, presenting experimental results and analysis. Section 4 provides the conclusions of this study.

## II. METHODOLOGY

### A. Feature extraction

The data-driven health indicators are derived from the datasets. All health indicators come from NASA and CALCE datasets. These extracted simple indicators and their aging performance are shown in Table I.

These health indicators are multidimensional features, each with varying degrees of correlation to the SOH. Including all health indicators in the output could introduce noise from less relevant features, thereby reducing prediction accuracy. Therefore, we eliminate low-correlation indicators and select high-correlation indicators for input into the network. In this research, We use Pearson correlation analysis to select the indices, and select the first five as the network input. Pearson correlation coefficient is of commonly used in the analysis of the relationship between SOH and health factors [34], and its calculation principle is shown in Eq (2).

$$A = \frac{\sum_{i=1}^{n}(a_i - \bar{a})(b_i - \bar{b})}{\sqrt{\sum_{i=1}^{n}(a_i - \bar{a})^2}\sqrt{\sum_{i=1}^{n}(b_i - \bar{b})^2}} \quad (2)$$

where $a_i$ and $b_i$ represent the values of the data points, with $\bar{a}$ and $\bar{b}$ denoting their respective mean values, and $n$ being the total number of data points.The Pearson correlation coefficient$A$ is an important measure of the linear correlation between two or more variables in statistics. If the value of the correlation coefficient is closer to 1, the correlation between the two variables is very significant, that is, there is a high degree of positive correlation between the two variables. The coefficients provide an intuitive way to explain complex correlations in data and help researchers and decision makers understand the potential relationship and impact between the two variables.

TABLE I: Details of Health Indicators (HIs)

| Abbreviation | Explanation | Aging Behavior |
|---|---|---|
| CCT | Constant current charging time | Shortens as battery ages due to increased internal resistance causing faster voltage rise |
| CVT | Constant voltage charging time | Lengthens as battery ages, with reduced current acceptance near full charge, lowering charging efficiency |
| DT | Discharge time | Shortens as battery ages, with increased internal resistance causing faster voltage drop |
| TT | Time to reach maximum temperature | Shortens as battery ages, with increased internal resistance generating more heat, causing faster temperature rise |
| R | Internal resistance | Increases as battery ages |
| CMT | Time for constant voltage charging current to drop to 1.5A | Shortens as battery ages, with decreased capacity and increased internal resistance causing faster current drop |
| CVI mean | Mean constant voltage charging current | Decreases as battery ages, with increased internal resistance and current dropping to a lower level until fully charged |
| CVI std | Standard deviation of constant voltage charging current | Increases as battery ages, with greater fluctuation |
| CCV mean | Mean constant current charging voltage | Decreases due to increased voltage drop from higher internal resistance |
| CCV std | Standard deviation of constant current charging voltage | Increases as battery ages, with greater fluctuation |
| CDV mean | Mean constant current discharging voltage | Decreases due to increased voltage drop from higher internal resistance |
| CDV std | Standard deviation of constant current discharging voltage | Increases as battery ages, with greater fluctuation |

### B. DAE-CNN-BiLSTM-Attention model

The raw input data is often noisy, especially during charge and discharge cycles. In most approaches, models input the raw data without denoising directly into the neural network, which significantly impacts the prediction accuracy. The algorithm firstly de-noises the training samples, and then injects the training samples into the deep neural network to ensure the stability and robustness of the algorithm. In this project, we propose to use the denoising self-encoder (DAE) to reconstruct the low-dimensional data by unsupervised learning and keeping as much information as possible [35].

In the last few years,attention mechanisms have been shown great promise in various deep learning tasks [36]. In this paper, we calculate attention scores using an attention mechanism, convert them into weights with the softmax function, and then apply these weights to the outputs of the LSTM to obtain context vectors. This approach is simple to implement,

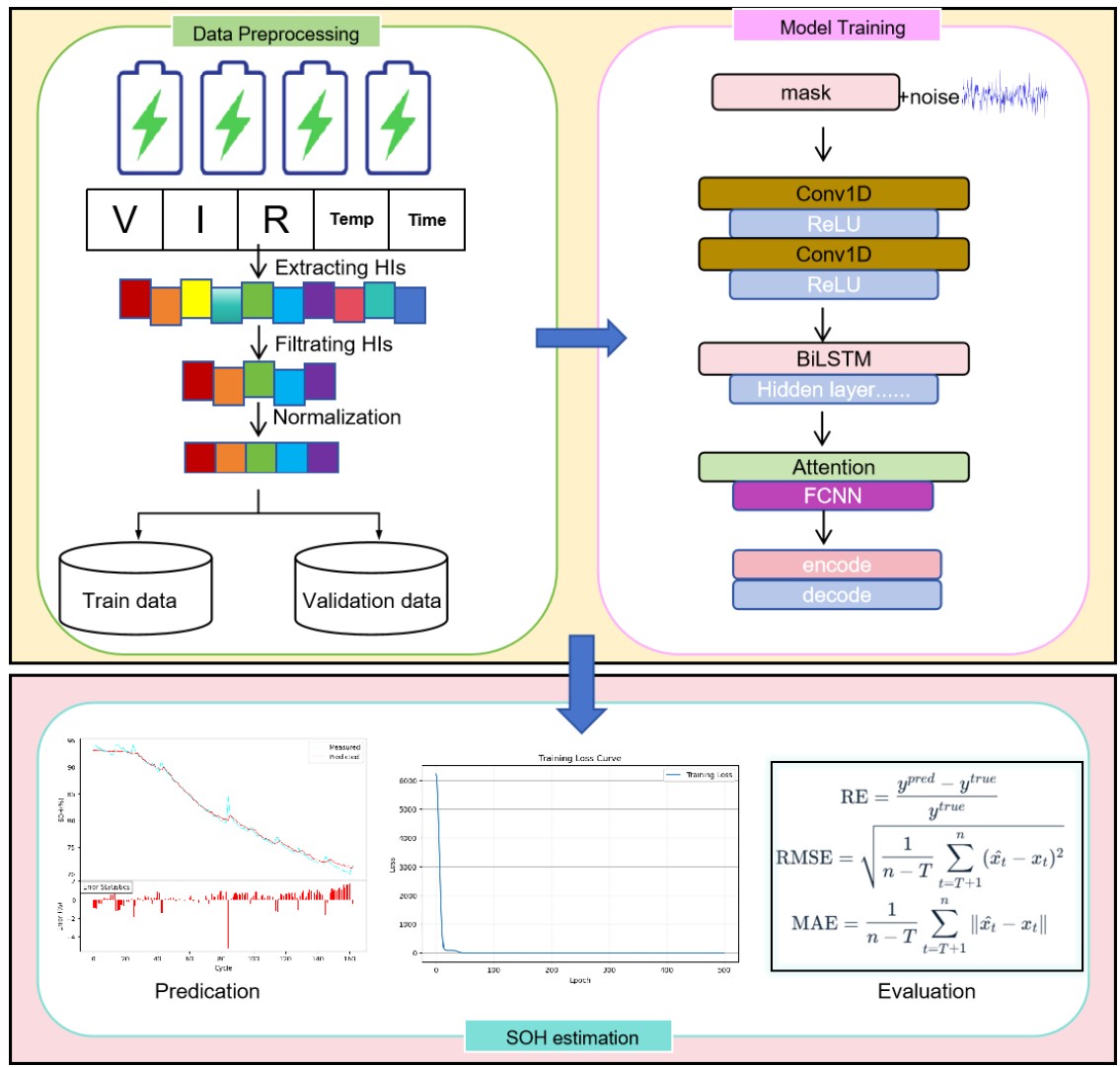

Fig. 1: Overall framwork of the proposed battery SOH estimation model.

computationally efficient, and well-suited for handling time-series data, highlighting important temporal information within the network.

Fig. 1. illustrates the framework of the DAE-CNN-BiLSTM-Attention model for predicting battery SOH, including denoising functionality and the CNN, BiLSTM, and Attention modules. In part A, health indicators are extracted, and the top 5 features are selected based on their Pearson correlation coefficients. These features are then normalized, with 70% used as training data and 30% as validation data. Gaussian noise is added, followed by two CNN layers to extract local features. The BiLSTM captures long-term dependencies in the sequential data, while the attention mechanism helps the network focus on the most important parts for predicting SOH. An autoencoder is employed to denoise the data by attempting to reconstruct the original data from the noisy input, thereby enhancing the robustness of the network.The model structures are summarized in Table II. The loss function converges to zero, and the model's performance is quantified using RMSE and MAE metrics.

## III. EXPERIMENT RESULTS AND ANALYSIS

### A. Datasets

The data from the NASA repository was collected by the NASA Ames Prognostics Center of Excellence (PCoE) on the NASA prognostics tested [11]. NASA batteries were used to validate the proposed method [35].This study utilizes batteries B0005, B0006, B0007,and B0018,abbreviations are used in the table, as B5,B6,B7,B18 respectively,Table III is the experimental conditions for these batteries.Fig. 2 shows the capacity degradation process of the NASA battery dataset.These batteries have a failure threshold of 1.4 Ah.

The CALCE dataset is a battery cycling test dataset from the Center for Advanced Life Cycle Engineering (CALCE) at the University of Maryland. CALCE batteries are widely used in battery state estimation studies and were used to validate the proposed method in [33]. This study uses batteries CS2_35, CS2_36, CS_37,and CS2_38,abbreviations are used in the

TABLE II: Neural network structure and parameters.

| Model | Structure | Number of Sampling Points |
|---|---|---|
| CNN | noisy input$\rightarrow X$
Conv1D(Channel: 64/Kernel: 3)$\rightarrow$ReLU$\rightarrow$
Conv1D(Channel: 128/Kernel: 3)$\rightarrow$ReLU$\rightarrow$ | X
64
128 |
| BiLSTM | Number of bidirectional layers: 1
Hidden_size: 100 $\rightarrow$ Hidden_size * 2 | 128
200 |
| Attention | Hidden_size * 2 $\rightarrow$
Attention_size: 20 $\rightarrow$
Fc(200$\rightarrow$1) | 20
200
1 |
| Encoder | encoder_fc1: input_size * sequence_length
$\rightarrow$ hidden_size
decoder_fc2: $\rightarrow$ input_size * sequence_length | 100
X
X |

TABLE III: Experimental conditions for NASA dataset.

| Battery | | B5 | B6 | B7 | B18 |
|---|---|---|---|---|---|
| | Normial capacity(Ah) | 2 | 2 | 2 | 2 |
| | Data length | 168 | 168 | 168 | 133 |
| | Ambient temperature(°C) | 24 | 24 | 24 | 24 |
| Charge | CC(A) | 1.5 | 1.5 | 1.5 | 1.5 |
| | cut-off current(mA) | 20 | 20 | 20 | 20 |
| | CV(V) | 4.2 | 4.2 | 4.2 | 4.2 |
| Discharge | CD(A) | 2 | 2 | 2 | 2 |
| | cut-off voltage(V) | 2.7 | 2.5 | 2.2 | 2.5 |

TABLE IV: Experimental conditions for calce dataset

| Battery | | C35 | C36 | C37 | C38 |
|---|---|---|---|---|---|
| | Normial capacity(Ah) | 1.1 | 1.1 | 1.1 | 1.1 |
| | Data length | 882 | 936 | 969 | 996 |
| | Ambient temperature(°C) | 1 | 1 | 1 | 1 |
| Charge | CC(A) | 0.5 | 0.5 | 0.5 | 0.5 |
| | cut-off current(mA) | 20 | 20 | 20 | 20 |
| | CV(V) | 4.2 | 4.2 | 4.2 | 4.2 |
| Discharge | CD(A) | 1 | 1 | 1 | 1 |
| | cut-off voltage(V) | 2.7 | 2.7 | 2.7 | 2.7 |

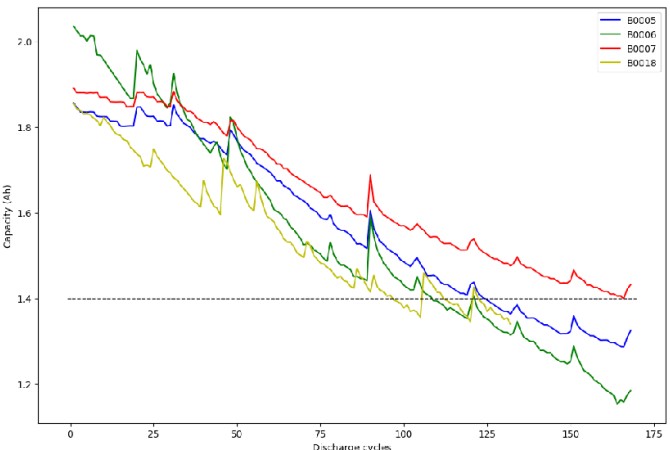

Fig. 2: NASA dataset capacity degration at ambient temperature of 24°C.

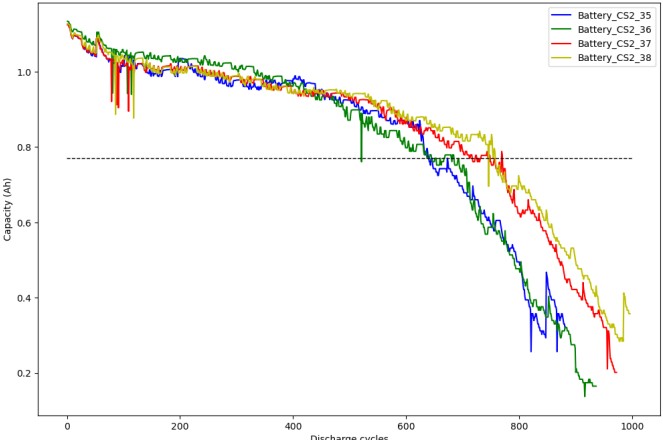

Fig. 3: CALCE dataset capacity degration at ambient temperature of 1°C.

table, as C35,C36,C37,C38 respectively,Table IV is the experimental conditions for these batteries.Fig. 3 shows the capacity degradation process of the CALCE battery dataset.These batteries have a failure threshold of 0.77 Ah.

### B. Feature Selection

As shown in table V, the top 5 health indicators are used as inputs to the model. For instance, the inputs selected for B0005 are 'CCT', 'DT', 'TT', 'CMT','CDV mean'. Fig. 4 , Fig. 5 illustrate the correlations between the various health indicators respectively,with positive correlation in red and negative correlation in blue.

### C. Overall performance

This study employs three commonly used metrics to quantify the performance of the model in predicting battery health status: Mean Absolute Error (MAE), Root Mean Squared Error (RMSE). The definitions of these metrics are as follows:

$$\text{MAE} = \frac{1}{n-T} \sum_{t=T+1}^{n} \|\hat{x}_t - x_t\| \qquad (3)$$

$$\text{RMSE} = \sqrt{\frac{1}{n-T} \sum_{t=T+1}^{n} (\hat{x}_t - x_t)^2} \qquad (4)$$

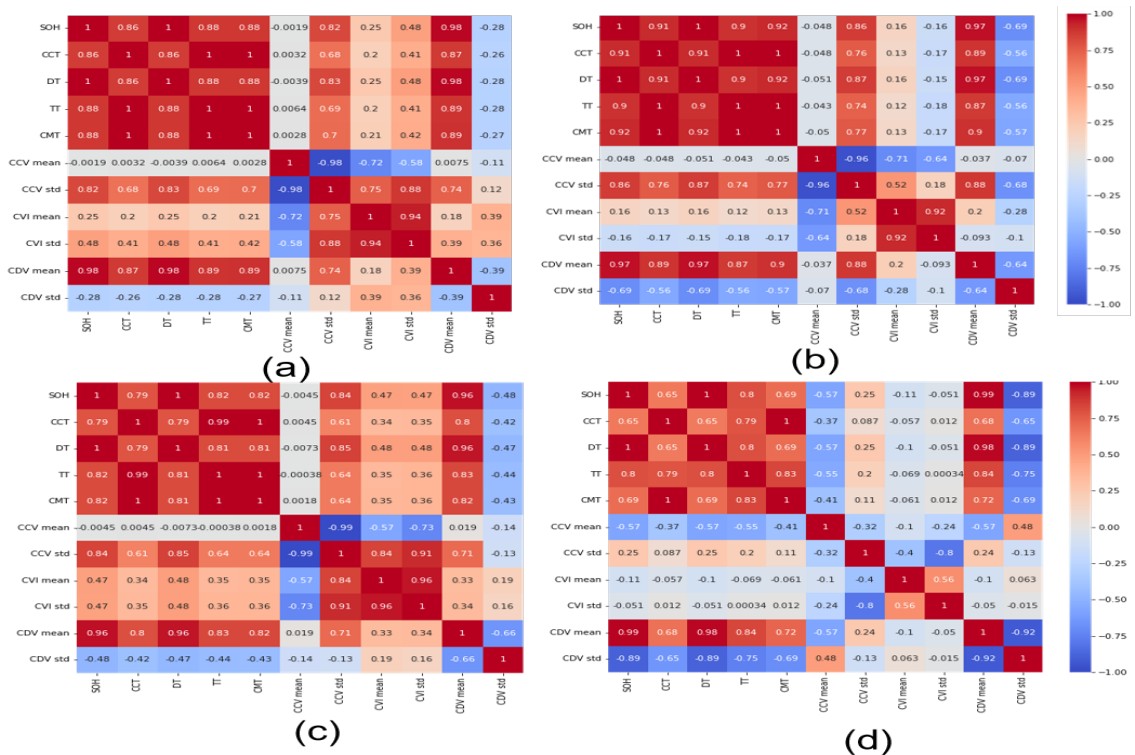

Fig. 4: NASA Pearson Correlation Heatmap:(a)B5;(b)B6;(c)B7;(d)B18.

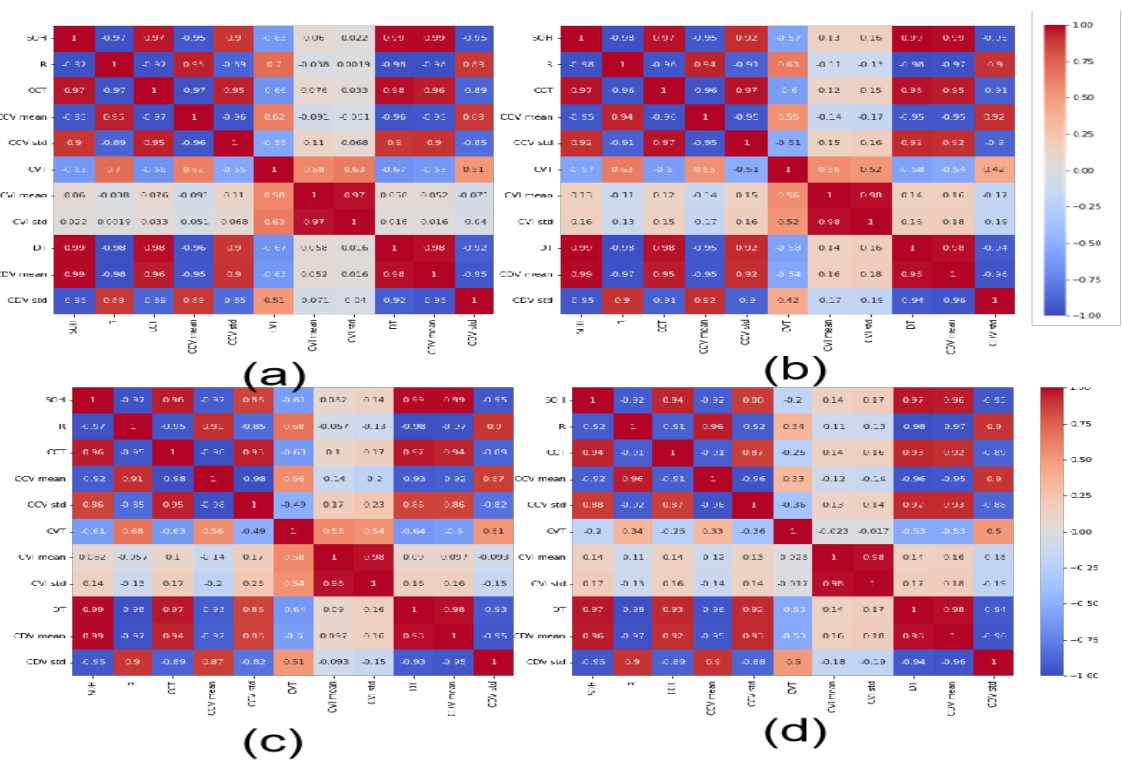

Fig. 5: CALCE Pearson Correlation Heatmap:(a)C35;(b)C36;(c)C37;(d)C38.

TABLE V: Correlation Coefficients of Health Indicators for Different Batteries.

| B0005 | B0006 | B0007 | B0018 |
|---|---|---|---|
| CCT 0.862586 | CCT 0.908604 | CCT 0.786684 | CCT 0.648654 |
| DT 0.999947 | DT 0.999915 | DT 0.999725 | DT 0.999773 |
| TT 0.877558 | TT 0.897890 | TT 0.816621 | TT 0.804534 |
| CMT 0.881882 | CMT 0.919521 | CMT 0.815183 | CMT 0.694932 |
| CCV mean -0.001942 | CCV mean -0.047989 | CCV mean -0.004465 | CCV mean -0.570699 |
| CCV std 0.822106 | CCV std 0.862932 | CCV std 0.839493 | CCV std 0.251537 |
| CVI mean 0.245934 | CVI mean 0.160218 | CVI mean 0.471227 | CVI mean -0.106422 |
| CVI std 0.475350 | CVI std -0.156089 | CVI std 0.468511 | CVI std -0.051191 |
| CDV mean 0.982357 | CDV mean 0.965189 | CDV mean 0.961071 | CDV mean 0.985401 |
| CDV std -0.283559 | CDV std -0.694572 | CDV std -0.482625 | CDV std -0.892615 |

| CS2_35 | CS2_36 | CS2_37 | CS2_38 |
|---|---|---|---|
| R -0.969031 | R -0.975631 | R -0.968516 | R -0.922052 |
| CCT 0.967323 | CCT 0.969335 | CCT 0.955910 | CCT 0.942224 |
| CCV mean -0.952510 | CCV mean -0.951288 | CCV mean -0.922861 | CCV mean -0.921456 |
| CCV std 0.897771 | CCV std 0.917804 | CCV std 0.855091 | CCV std 0.875672 |
| CVT -0.626522 | CVT -0.565320 | CVT -0.612713 | CVT -0.197397 |
| CVI mean 0.060142 | CVI mean 0.133853 | CVI mean 0.081966 | CVI mean 0.142492 |
| CVI std 0.022159 | CVI std 0.156692 | CVI std 0.144904 | CVI std 0.173008 |
| DT 0.991876 | DT 0.994180 | DT 0.991499 | DT 0.967231 |
| CDV mean 0.990909 | CDV mean 0.989263 | CDV mean 0.988342 | CDV mean 0.955991 |
| CDV std -0.947330 | CDV std -0.949600 | CDV std -0.945006 | CDV std -0.952266 |

TABLE VI: SOH estimate MAEs and RMSEs on NASA and CALCE datasets.

| Datasets | Metrics | LSTM | At-LSTM | CNN-BiLSTM | CNN-BiLSTM-At | DAE-CNN-BiLSTM-At |
|---|---|---|---|---|---|---|
| B0005 | MAE | 1.0882 | 1.0880 | 0.8882 | 0.5521 | **0.5075** |
| | RMSE | 1.5428 | 1.3567 | 1.3393 | 0.7334 | **0.7064** |
| B0006 | MAE | 1.6684 | 1.2518 | 1.1133 | 1.2459 | **0.8462** |
| | RMSE | 2.2141 | 1.7345 | 1.6578 | 1.2806 | **1.2405** |
| B0007 | MAE | 1.1695 | 1.1872 | 0.9839 | 0.5992 | **0.4407** |
| | RMSE | 1.3116 | 1.5503 | 1.4796 | 0.9050 | **0.6337** |
| B0018 | MAE | 1.4277 | 1.2273 | 0.9233 | 0.7266 | **0.7258** |
| | RMSE | 1.8202 | 1.8140 | 1.2973 | 1.1352 | **0.9738** |
| CS2_35 | MAE | 0.0488 | 0.0470 | 0.0485 | 0.0478 | **0.0154** |
| | RMSE | 0.0267 | 0.0294 | 0.0228 | 0.0197 | **0.0191** |
| CS2_36 | MAE | 0.0373 | 0.0382 | 0.0341 | 0.0337 | **0.0266** |
| | RMSE | 0.0391 | 0.0341 | 0.2546 | **0.0230** | 0.0303 |
| CS2_37 | MAE | 0.0315 | 0.0226 | 0.0335 | 0.0371 | **0.0207** |
| | RMSE | 0.0571 | 0.0262 | 0.0380 | 0.0364 | **0.0335** |
| CS2_38 | MAE | 0.0384 | 0.0358 | 0.0261 | **0.0227** | 0.0286 |
| | RMSE | 0.0511 | 0.0498 | 0.0522 | 0.0713 | **0.0509** |

Where $C_n$ represents the length of the sequence, and $C_T$ represents the length of the training sequence samples. MAE (Mean Absolute Error) is is a method of calculating the average absolute error between measured and predicted values, which measures the average difference between them. RMSE (Root Mean Square Error) is the mean difference between the predicted value and actual values, providing the standard deviation of the errors.

We designed several experiments to validate the performance of the proposed model. table VI presents the evaluation results, with the best results highlighted in bold. B0005 has MAE 0.5075 and RMSE 0.7064, B0006 has MAE 0.8462 and RMSE 1.2405, B0007 has MAE 0.4407 and RMSE 0.6337, B0018 has MAE 0.7258 and RMSE 0.9738, CS2_ 35 has MAE 0.0154 and RMSE 0.0191, CS2_36 has MAE 0.0266 and RMSE 0.0303, CS2_37 has MAE 0.0207 and RMSE 0.0335, CS2_38 has MAE 0.0028 and RMSE 0.0509, among which, except the RMSE of CS2_36 is lower than the 0.0230 of CNN-BiLSTM-At, and the MAE of CS2_38 is lower than the 0.0227 of CNN-BiLSTM-At, other indicators show the best. The results of denoising and non-denoising are compared, which indicates that there is still room for improvement of our denoising function The DAE-CNN-BiLSTM-Attention model consistently shows the smallest MAE and RMSE, demonstrat-

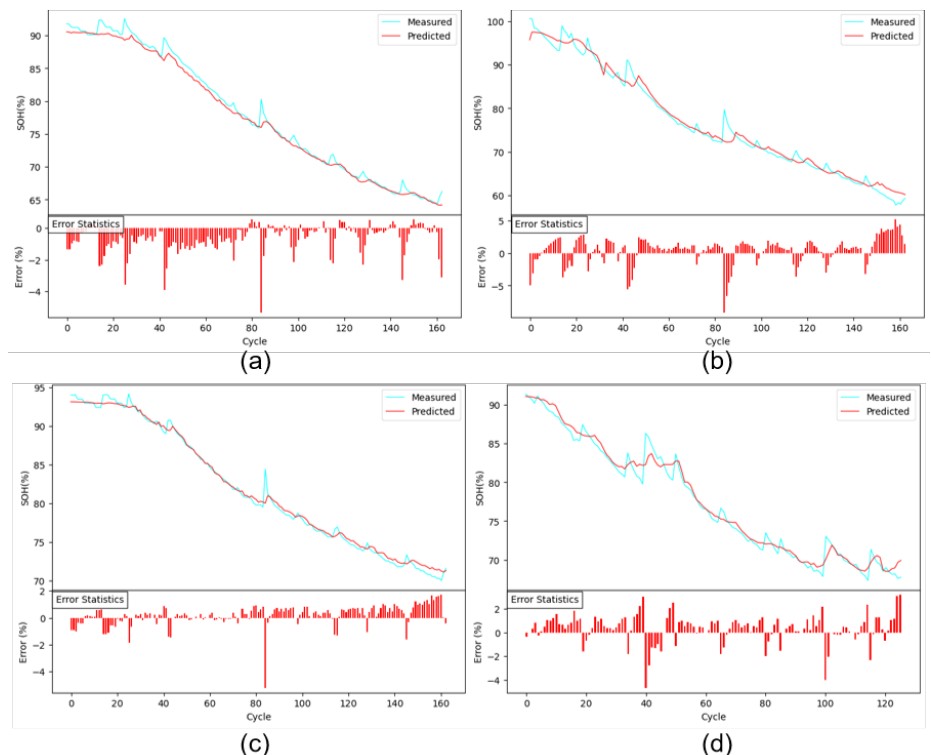

Fig. 6: NASA capacity estimation results and errors:(a)B5;(b)B6;(c)B7;(d)B18.

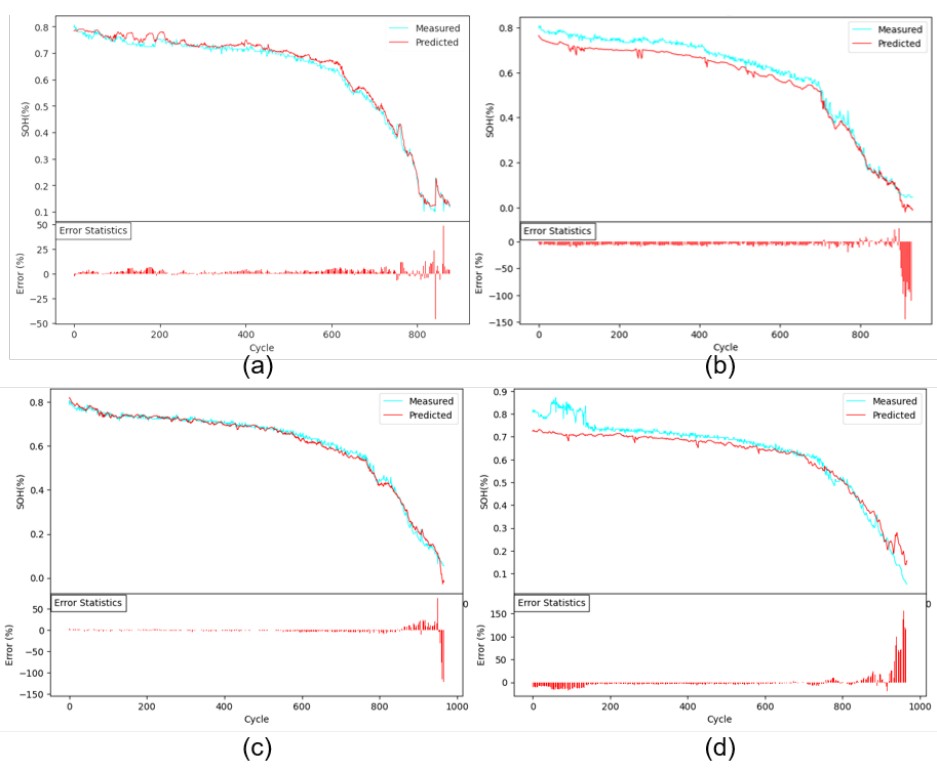

Fig. 7: CALCE capacity estimation results and errors:(a)C35;(b)CS36;(c)C37;(d)C38.

ing superior predictive performance. The best evaluation result for this model was achieved on the CS2_35 battery,Mean Absolute Error (MAE) is 0.0154 and a Root Mean Square Error (RMSE) is 0.0191. Compared to the model without denoising, the performance improvements in MAE and RMSE were 55.4% and 3.14%, respectively. Additionally, the model is simple, requiring only one minute to complete 500 training iterations, which is significantly faster compared to the 90 minutes and 10 minutes reported in paper [11].

Fig. 6 clearly shows the predictions of NASA battery results of state of health and their errors, which are close to the real battery health, with all errors within 5% even at peak anomaly points.Fig. 7 shows the prediction and error of the CALCE battery health state.This study utilized 70% of the battery data for training and predicted the entire degradation process. Compared to the NASA dataset, the CALCE dataset has a significantly larger data volume and more anomaly noise, which increases the difficulty of prediction. Although the error has increased somewhat, it still remains close to the actual degradation curve.

## IV. CONCLUSION

Accurately estimating the State of Health (SOH) of batteries is critical for effective battery management, and establishing a reliable prediction network is key. We propose a data-driven hybrid neural network for SOH prediction. Initially, we extract over ten features from the batteries and select the top five features based on their absolute Pearson correlation coefficients for input into the network. The Convolutional Neural Network (CNN) first extracts features from the noisy input data, then the Bidirectional Long Short-Term Memory (BiLSTM) network learns the degradation information of the battery, the Attention mechanism focuses on important information, and finally, the autoencoder-decoder restores the noisy data to its original state, enhancing the model's adaptability and stability. The proposed model was validated on different battery datasets and demonstrated lower Mean Absolute Error (MAE) and Root Mean Square Error (RMSE) compared to other models

In the future, we will make improvements in the following three directions:

1. Make more improvements to the denoising function to improve its performance.

2. Focus on the spatial information between features.

3. Compared with more single classical models, find the advantages and disadvantages of each model to improve the accuracy of prediction.

## ACKNOWLEDGMENT

The authors would like to thank Teacher Huang for his encouragement and support.

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
