# OpenReview forum: "A Data-Driven DAE-CNN-BiLSTM-Attention Prediction Model for the State of Health of Lithium-ion Batteries"
_IEEE.org/ICIST/2024/Conference — IEEE ICIST 2024 Conference Submission_

### Official Review · Reviewer_79Wn · 2024-08-21
**This article is quite fascinating and of high quality.**

**Rating:** 7
**Confidence:** 3

**Review:**

The paper titled "A Data-Driven DAE-CNN-BiLSTM-Attention Prediction Model for the State of Health of Lithium-ion Batteries" proposes a hybrid model: DAE-CNN-BiLSTM-Attention to achieve accurate health state predictions. The model integrates the advantages of local feature extraction from convolutional neural networks, temporal dependency learning from bidirectional long short-term memory networks (BiLSTM), effective weight allocation through attention mechanisms, and the noise reduction capabilities of denoising autoencoders (DAE) to enable the network to better adapt to complex real-world environments. My specific feedback is as follows: 1) The research challenges and innovations of this paper can be further explained in this paper. Please add these details. 2) Some formatting issues need to be addressed.

---

### Official Review · Reviewer_NhRp · 2024-08-21
**Accept**

**Rating:** 7
**Confidence:** 3

**Review:**

This paper proposed a data-driven hybrid neural network for SOH prediction. The theory is correct and can be accepted after responding the following comments.
(1)	What is the contribution of the paper? It should be highlighted both in the introduction and in the content.
(2)	It can be compared with existing articles to make the innovation point clearer.
(3)	please check if you need to update your Introduction/Related Work section to include latest closely relevant references that have appeared in journals and/or conferences in the past two years.

---

### Official Review · Reviewer_z3nz · 2024-08-22
**This article is very interesting and a good one**

**Rating:** 7
**Confidence:** 3

**Review:**

This study extracted over ten health indicators and designed a hybrid model: DAE-CNN-BiLSTM-Attention , which demonstrated excellent performance. The theory is correct and can be accepted after responding the following comments.
(1) In the introduction, it is not enough to state the current work. It should be expended and reconstructed.
(2) In the simulation section, more analysis can be added to better explain the main results of this paper, that's not enough.
(3) There are many typos and grammar errors. The authors should have a native English speaker or software packages to perform the editing check.
(4) The conclusion of the article suggests using the present perfect tense for description.

---

### Comment · Reviewer_z3nz · 2024-08-21
**This article is very interesting and a good one**

This study extracted over ten health indicators and designed a hybrid model: DAE-CNN-BiLSTM-Attention , which demonstrated excellent performance. The theory is correct and can be accepted after responding the following comments.
(1) In the introduction, it is not enough to state the current work. It should be expended and reconstructed.
(2) In the simulation section, more analysis can be added to better explain the main results of this paper, that's not enough.
(3) There are many typos and grammar errors. The authors should have a native English speaker or software packages to perform the editing check.
(4) The conclusion of the article suggests using the present perfect tense for description.

---

### Decision · Program_Chairs · 2024-09-06

Accept (Oral)